# Uncertainty Quantification for Ti-7Al Alloy Microstructure with an Inverse Analytical Model (AUQLin)

**DOI:** 10.3390/ma12111773

**Published:** 2019-05-31

**Authors:** Pınar Acar

**Affiliations:** Department of Mechanical Engineering, Virginia Polytechnic Institute and State University, Blacksburg, VA 24061, USA; pacar@vt.edu

**Keywords:** uncertainty quantification, microstructure, multi-scale modeling, titanium

## Abstract

The present study addresses an inverse problem for observing the microstructural stochasticity given the variations in the macro-scale material properties by developing an analytical uncertainty quantification (UQ) model called AUQLin. The uncertainty in the material property is modeled with the analytical algorithm, and then the uncertainty propagation to the microstructure is solved with an inverse problem that utilizes the transformation of random variables principle. The inverse problem leads to an underdetermined linear system, and thus produces multiple solutions to the statistical features of the microstructure. The final solution is decided by solving an optimization problem which aims to minimize the difference between the computed and experimental statistical parameters of the microstructure. The final result for the computed microstructural uncertainty is found to provide a good match to the experimental microstructure information.

## 1. Introduction

Uncertainty in thermo-mechanical processing conditions is unavoidable, as it originates from temperature and stress gradients. The processing uncertainty propagates through the microstructure and material properties, and impacts the expected performance of engineering components. With the introduction of the Integrated Computational Materials Engineering (ICME) paradigm [1], a considerable amount of effort has been dedicated to computational models that aim to model the relationship between material response and the underlying microstructure. However, the computational models are mostly based on deterministic microstructural variables, and hence fail to model the uncertainty and its effects on the expected material response. On the other hand, the propagation of the processing uncertainty can be unpredictable due to the complex relationships between the processing, microstructure, and material properties. In some cases, it can lead to significant inefficiency in properties and cause partial or complete failure of the engineering components. Therefore, uncertainty quantification (UQ) in computational materials modeling has been recognized as an important future topic. The state-of-the-art UQ models mostly employ computationally expensive techniques including Monte Carlo Simulation [2,3,4,5,6,7] and thus are not well-suited for multi-scale modeling and design problems. Such numerical techniques are not feasible for the 3D multi-scale modeling of polycrystals, where the computational time requirement for a single simulation can range from hours to days [8] depending on the resolution of the macro- and meso-scale finite element meshes. In the literature, the effects of the uncertainty have been studied for nonlinear models [9,10,11], which can be of interest to observe the uncertainty in nonlinear multi-scale material systems. However, given the computationally expensive nature of the multi-scale problem of interest for a potential future work, one powerful approach has been found to model the stochasticity of the microstructure by implementing an analytical solution framework. For this purpose, an analytical linear solution scheme was developed by the author and has been applied to model the propagation of the uncertainty by assuming that the microstructural variables follow a Gaussian distribution [12,13,14,15]. The analytical algorithm has also been exercised in multi-scale design problems [14,15] such that for a prescribed variation in a volume-averaged material property, the variations in the microstructure are solved and considered in optimizing the material response. However, some developments are required for the analytical algorithm. First, with its previous form, it can model the variations which only fit into Gaussian distribution. Second, the results of the analytical algorithm have not been compared to the experimental data for inverse problems that aim to capture the microstructural stochasticity given the variations in the volume-averaged properties. Therefore, in the present work, our main goals were: (i) to achieve a more generalized analytical algorithm which can model not only the Gaussian distribution but also the variations in the experimental data that do not fit into the Gaussian representation; (ii) to achieve an analytical UQ model that can capture the variations in the experimental data for the Titanium–aluminum alloy Ti-7Al, microstructure, and Young’s modulus value. The generalized analytical UQ algorithm will utilize linear programming (LP), and thus from now on it will be called the “analytical UQ with LP” model, or the “AUQLin” model in short. The development of the generalized AUQLin model for the presented meso-scale uncertainty problem is a first step towards its integration with the computationally expensive 3D multi-scale modeling and design problems in future to capture the propagation of the microstructural uncertainty across different length scales.

The microstructure modeling in the present work is based on the use of the orientation distribution function (ODF). The ODF measures the volumes of different crystallographic orientations in a polycrystalline material. The ODF can be defined with parameterization techniques such as Euler angles [16] and Rodrigues parameters [12,17]. In this work, we use a local finite element discretization of the ODF values [12,13] using Rodrigues orientation space. The local discretization of the ODF with the finite element method has been found to be a desirable approach owing to the ease of modeling sharp textures, including single crystals. Therefore, it gives flexibility to model different texture components, which would be a handful feature to analyze the uncertainty in the microstructural texture. The uncertainty modeling in this work starts with modeling the variations in the Young’s modulus of Ti-7Al using the available experimental data. Choosing an elastic parameter would be helpful to understand the uncertainty in the initial microstructure. This is because Young’s modulus data do not contain any other uncertainty associated with the plastic behavior of the material. The AUQLin model is utilized to quantify the uncertainty in the Young’s modulus of Ti-7Al alloy and model its propagation to the microstructure by solving an inverse problem with LP. The statistical data of the Ti-7Al microstructure are also experimentally available through the electron backscatter diffraction (EBSD) samples. Because of the definition of the ODF approach, there is a direct transformation between the EBSD data and the ODF. This is because each color in an EBSD sample shows a different crystallographic orientation, whereas the ODF is defined as the volume fraction of different orientations (or colors in the EBSD image). Therefore, the uncertainty in the experimental ODF values is observed with this direct transformation feature. Finally, we solve an optimization problem to minimize the difference between the computed and experimental statistical data of the ODF values. The variations in the alloy microstructure obtained with the AUQLin model are found to match well with the microstructural uncertainty captured by the EBSD samples.

The remainder of this paper is organized as follows. In Section 2, the microstructure model is introduced. Section 3 addresses the AUQLin model and the inverse solution approach. Section 4 illustrates the results that are obtained for the microstructure uncertainty and provides a discussion of the approach and results. The summary of the paper and potential extensions as future work are given in Section 5.

## 2. Microstructure Modeling

The microstructural texture was modeled using the one-point probabilistic descriptor, ODF. The ODF, denoted by *A*, measures the volumes of different crystallographic orientations. The orientation spaces of the polycrystalline materials can be reduced to a smaller subset, which is known as the fundamental region after applying the crystal symmetries. Each unique orientation in the fundamental region is associated with a coordinate, *r*. In this work, the microstructure is represented using the Rodrigues parameterization. The fundamental region is discretized using a local finite element scheme with *N* independent nodes and $N_{elem}$ finite elements (and $N_{int}$ integration points per element). The discretization over the Rodrigues orientation space is advantageous, as the Rodrigues space leads to regular geometries. In this local finite element representation, each nodal point of the finite element mesh is associated with an ODF value, which is related to the volume of the corresponding crystallographic orientation. This compact representation enables the computation of volume-averaged material properties of microstructures using the nodal point ODF values. First, the ODF must satisfy the unit volume fraction constraint over the fundamental region, which is defined as follows: (1)$$\int_{R}Adv=\sum_{n=1}^{N_{elem}}\sum_{m=1}^{N_{int}}A\left(r_{m}\right)w_{m}\left|J_{n}\right|\frac{1}{{\left(1+r_{m}\cdot r_{m}\right)}^{2}}=1.$$

In Equation (Equation 1), $A(r_{m})$ is the ODF value at the $m^{th}$ integration point (that has a coordinate $r_{m}$), $|J_{n}|$ is the Jacobian value of the $n^{th}$ element, $w_{m}$ is the integration weight for the $m^{th}$ integration point, and $\frac{1}{{\left(1+r_{m}\cdot r_{m}\right)}^{2}}$ is the metric of the Rodrigues parameterization. The summation expression in Equation (Equation 1) can be written as a linear equation in terms of the nodal point ODF values [12,13,14,15]: (2)$$q^{T}A=1,$$
where *q* is the volume normalization vector that includes the single-crystal values. Equation (Equation 2) defines a mathematical constraint that must always be satisfied when modeling with the ODF approach. This constraint mathematically means that the sum of the volume fractions of different orientations over the fundamental region must be equal to one. Using a similar approach, any orientation-dependent volume-averaged material property, <χ>, can also be computed using single-crystal property values and the nodal point ODFs: (3)$$<\chi >=\int_{R}\chi \left(r\right)A\left(r\right)dv=\sum_{n=1}^{N_{elem}}\sum_{m=1}^{N_{int}}\chi \left(r_{m}\right)A\left(r_{m}\right)w_{m}\left|J_{n}\right|\frac{1}{{\left(1+r_{m}\cdot r_{m}\right)}^{2}}.$$

Equation (Equation 3) can also be written in the linear form in terms of the nodal points ODFs such that [12,13,14,15]: (4)$$<\chi >=p^{T}A,$$
where *p* is the matrix of the single-crystal values of the corresponding material property. More details about the Rodrigues parameterization and finite element discretization schemes can be found in our earlier works [12,13,14,15], and are not repeated here for brevity. In this work, the hexagonal close-packed (HCP) Ti-7Al microstructure was modeled with 50 independent ODF values, and the finite element representation of the fundamental region is shown in Figure 1.

In the present work, our goal was to understand the uncertainty in the microstructural texture using the measured uncertainty in the Young’s modulus value of the Ti-7Al alloy microstructure. To compute the Young’s modulus using the ODF values that represent the microstructural texture, we first computed the compliance values, *S*, using lower-bound averaging (Reuss average). After computation of the compliance values, the measured Young’s modulus value, $E_{11}$, could be computed as: $E_{11}=\frac{1}{S_{11}}$. The polycrystal compliance parameter, $S_{11}$, could be computed in terms of the ODF values using the volume-averaging equation (Equation (Equation 4)) as shown in Equation (Equation 5):(5)$$S_{11}=p_{11}^{T}A,$$
where $p_{11}$ are the single-crystal values for the $S_{11}$ as represented in Figure 2:

## 3. Uncertainty Quantification in Material Properties

The uncertainty in the microstructural texture was modeled through the variations in the ODF values. The measured uncertainty in Young’s modulus value of the Ti-7Al was used to find the texture uncertainty. The uncertainty in the ODF values was found with an inverse scheme by first finding the uncertainty in the compliance parameters. The measurements for Young’s modulus showed that its value can vary around 119±3 MPa for a randomly oriented microstructural texture, as indicated by the tensile and compressive stress–strain curves of Ti-7Al presented in our previous works [18,19]. We assumed that the variations in Young’s modulus value agreed with a uniform distribution on the given [116,122] MPa interval. The choice of probability distribution to model the variations in Young’s modulus affects the distributions of compliance and ODF values. For example, the use of a Gaussian distribution assumption for Young’s modulus would lead to a bimodal distribution for the compliance parameters because of the inverse relationship between Young’s modulus, $E_{11}$, and the compliance parameter, $S_{11}$. Since the relationship between the ODFs and compliance value, $S_{11}$, is defined in terms of the linear volume-averaging equation (Equation (Equation 5)), the probability distribution of the ODFs would be the same with the distribution of the compliance parameter, $S_{11}$. However, the bimodal distribution of the ODF would not reflect the expected uncertainty for the randomly oriented texture. This is because the bimodal distribution typically has two different peak values. However, it has already been shown that the experimental texture has variations around the random ODF values [12,13]. Therefore, the second peak in the probability distribution of the ODFs cannot be possible. On the other hand, the uniform distribution assumption for Young’s modulus, $E_{11}$, leads to an inverse uniform distribution for the compliance, $S_{11}$, and for the ODFs with a narrow support space. The narrow support region actually means that the minimum and maximum values that any ODF can take would be in the close vicinity of the random ODF value, and thus it would agree with the previous findings in [12,13] about the experimental microstructural texture data. With the uniform distribution assumption for Young’s modulus, $E_{11}$, we first applied the transformation of random variables rule to identify the probability distribution of the compliance parameter, $S_{11}$, which has a nonlinear relationship with the input parameter $E_{11}$. The details on the transformation of random variables rule are discussed in the following.

### 3.1. Uncertainties in the Nonlinear Material Properties

Given a nonlinear relationship between the input and output parameters, the probability density function (PDF) of the output variable can still be computed using the PDF of the input variable. This is possible with the use of the transformation of random variables technique. To apply this method, the relationship between an input variable, *x*, and output variable, *y*, should be explicitly defined using y=h(x) and inverted as x=u(y). If these conditions are met, then a Jacobian is computed using this explicit relationship as J=du/dy. Next, the PDF of the output variable can be written as a product of the PDF of the input variable and the Jacobian value as shown in Equation (Equation 6):(6)$$f_{y}\left(y\right)=f_{x}\left[u\left(y\right)\right]\times \left|J\right|,$$
where $f_{x}$ and $f_{y}$ define the PDFs of input and output parameters respectively. The PDF of the output variable, $f_{y}$, is the only unknown in Equation (Equation 6) because of the known explicit relationship between *x* and *y*. Therefore, $f_{y}$ can be computed with the given expression in Equation (Equation 6). Next, the statistical quantities including the expected value and variance of the output variable, E[y] and $\sigma_{y}^{2}$ respectively, are computed using the given expressions in Equations (Equation 7) and (Equation 8):(7)$$E\left[y\right]=\int_{y_{min}}^{y_{max}}yf_{y}\left(y\right)dy,$$
(8)$$\sigma_{y}^{2}=E\left[{\left(y-E\left[y\right]\right)}^{2}\right]$$
where $y_{min}$ and $y_{max}$ are the minimum and maximum values that the output parameter, *y*, can have. The minimum and maximum values of the output parameter can be computed with the equation that defines the explicit relationship between the input and output, y=h(x). The approach is first demonstrated to compute the PDF of the homogenized compliance value. In this computation, the nonlinear relationship arises from the expression $S_{11}=\frac{1}{E_{11}}$. Given the uniform probability distribution assumption for $E_{11}$ within the support space, [116,122] MPa, the expected value and the variance of $S_{11}$ were computed using the transformation of random variables method, and found as follows:(9)$$E\left[S_{11}\right]=\frac{ln\left(E_{11}^{max}\right)-ln\left(E_{11}^{min}\right)}{E_{11}^{max}-E_{11}^{min}},$$
(10)$$\sigma_{S_{11}}^{2}=\left(\frac{1}{E_{11}^{min}\cdot E_{11}^{max}}\right)-{\left(\frac{ln\left(E_{11}^{max}\right)-ln\left(E_{11}^{min}\right)}{E_{11}^{max}-E_{11}^{min}}\right)}^{2},$$
where $E_{11}^{min}$ and $E_{11}^{max}$ show the minimum and maximum values of the Young’s modulus, which were determined as 116 and 122 MPa, respectively. Using these values, the expected value and variance of $S_{11}$ were calculated as: $E[S_{11}]=0.0084$ (1/MPa) and $\sigma_{S_{11}}^{2}=1.4974\times 10^{-8}$ (1/MPa${}^{2}$). Next, the uncertainty in the microstructural texture was found using the computed uncertainty in the compliance parameter, $S_{11}$.

### 3.2. Computation of the Uncertainty in the Microstructural Texture

The next step in the present work was to quantify the uncertainty in the microstructural texture using the computed uncertainty in the compliance parameter, $S_{11}$. As discussed previously, the $S_{11}$ parameter follows an inverse uniform distribution with a PDF computed with the transformation of random variables rule. The uncertainty in the microstructural texture is characterized by the variations in the ODF values. Since the explicit relationship between the ODF values and the compliance parameter ($S_{11}$) is known through Equation (Equation 5), the transformation of random variables rule can be applied to identify the PDF of the ODF values. After the identification of the PDF, the expected values and variances of the ODF values were also computed with the given formulations in Equations (Equation 7) and (Equation 8). Because of the linear relationship between the ODFs and compliance, the expected value and variance could be found by solving the following linear systems: (11)$$\mu_{S_{11}}=p_{11}^{T}\mu_{A},$$
(12)$$\sigma_{S_{11}}^{2}=p_{11}^{2}\left(i\right)\sigma_{A}^{2}\left(i\right),\;\;\mathrm{where}\;\;i=1,2,\ldots ,50,$$
where Equations (Equation 11) and (Equation 12) show the expected value and variance equations for the ODF values. Note that Equation (Equation 11) is written in terms of the mean values (where the ODF mean values are denoted by $\mu_{A}$ and the mean value of $S_{11}$ is shown with $\mu_{S_{11}}$) rather than using the expected value. Since the microstructural texture was modeled with 50 independent ODF values but the uncertainty in the texture was obtained using a single volume-averaged material property (which leads to two equations for the mean value and variance computation), an underdetermined linear system of equations should be solved to find the statistical features of the ODFs. The underdetermined linear system can produce multiple solutions. The variables solved in this underdetermined system are the mean values of the ODFs and the variances of the ODF values, which makes 100 variables to be solved. The ODF values were assumed to be independent, which means that the cross-covariance terms were assumed to be zero. This assumption was made in order to avoid further increasing the number of variables since the number of available equations (mean value and variance) would not chang with the increase in the number of variables and it would lead to more parametric solutions. The independent ODF assumption was also shown to lead to negligible errors when the results of the analytical solution based on the Gaussian assumption were compared to the experimental data of Ti-7Al in our previous works [12,13]. The variance expression in Equation (Equation 12) was derived with this assumption for the variance of the each ODF value ($\sigma_{A}^{2}\left(i\right)$, i=1,2,…50) using the property matrix elements for the compliance ($p_{11}\left(i\right)$
i=1,2,…50). With the consideration of the diagonal terms, the $p^{T}\Sigma_{A}p=0$ (where $\Sigma_{A}$ is the full ODF symmetric covariance matrix) is reduced to $p^{2}\sigma_{A}^{2}=0$.

## 4. Results and Discussion

The uncertainty in the ODF was calculated by solving the linear system of equations. In addition to the mean value and variance equations discussed previously (Equations (Equation 11) and (Equation 12)), two other mathematical equations must be defined to satisfy the unit volume fraction constraint (Equation (Equation 2)). This constraint is imposed on the system of equations in terms of a mean value equation and a variance expression. The unit volume normalization is a strict mathematical constraint. Therefore, the final solution obtained with the analytical method for the ODF values must satisfy this constraint, which is only possible by enforcing zero-variance condition. The mathematical expressions solved to find the mean values and variances of the ODFs are given in Equation (Equation 13): (13)$$\begin{array}{c} \begin{array}{c} p_{11}\left(1\right)\mu_{A}\left(1\right)+p_{11}\left(2\right)\mu_{A}\left(2\right)+\ldots +p_{11}\left(50\right)\mu_{A}\left(50\right)=\mu_{S_{11}},\\ p_{11}^{2}\left(1\right)\sigma_{A}^{2}\left(1\right)+p_{11}^{2}\left(2\right)\sigma_{A}^{2}\left(2\right)+\ldots +p_{11}^{2}\left(50\right)\sigma_{A}^{2}\left(50\right)=\sigma_{S_{11}}^{2},\\ q\left(1\right)\mu_{A}\left(1\right)+q\left(2\right)\mu_{A}\left(2\right)+\ldots +q\left(50\right)\mu_{A}\left(50\right)=1,\\ q^{2}\left(1\right)\sigma_{A}^{2}\left(1\right)+q^{2}\left(2\right)\sigma_{A}^{2}\left(2\right)+\ldots +q^{2}\left(50\right)\sigma_{A}^{2}\left(50\right)=0,\\ x={\left[\mu_{A}\left(1\right),\mu_{A}\left(2\right),\ldots ,\mu_{A}\left(50\right),\sigma_{A}^{2}\left(1\right),\sigma_{A}^{2}\left(2\right),\ldots ,\sigma_{A}^{2}\left(50\right)\right]}^{T}\;,\;\;\;\mathrm{where}\;\;x\geq 0, \end{array} \end{array}$$
where *x* defines the set of variables. The variables of this system must be non-negative in order to satisfy the ODF non-negativity condition (A≥0), and this leads to a non-negative mean values ($\mu_{A}\geq 0$) condition since the variances are already non-negative by definition. The covariance expression for the unit volume fraction constraint is enforced through the equations defined through $\Sigma_{A}q=0$ for each row of the ODF covariance matrix, $\Sigma_{A}$, as shown in the proof given in our previous work [14]. Equation (Equation 13) is solved as an LP problem. The standard definition of an LP problem is given as follows: (14)$$\begin{array}{c} \min\;f^{T}x\\ \mathrm{subject}\;\mathrm{to}:\;Ax\leq b\\ \mathrm{subject}\;\mathrm{to}:\;A_{eq}x=b_{eq}\\ \mathrm{where}\;lb\leq x\leq ub, \end{array}$$
where $f^{T}$ shows the coefficient of the objective function to be minimized, *A* and $A_{eq}$ are the coefficients of the inequality and equality constraints, respectively, and *b* and $b_{eq}$ are the right-hand-side values of the same constraint equations. The lower and upper bounds of the design variables are denoted by lb and ub, respectively. To understand the uncertainty in the ODF values, our goal was to obtain the mean values and the variances. The given problem definition in Equation (Equation 13) treats the mean values and variances as independent variables. Therefore, rather than solving the LP problem for both mean values and variances at the same time, the problem was divided into two subproblems to find the mean values and variances separately. Using the general LP problem given in Equation (Equation 14), the objective function in the first problem was defined using the mean value equation for the compliance parameter, $p_{11}^{T}\mu_{A}=\mu_{S_{11}}$, through an introduced error parameter, *r*, as follows: (15)$${p_{11}}^{T}\left(\mu_{A}+r\right)=\mu_{S_{11}}.$$

With the introduction of the error parameter, *r* in Equation (Equation 15), the objective was defined as the minimization of this error value, by minimizing $p_{11}^{T}r$. However, to avoid an unbounded linear solution, the objective function was defined in terms of a single variable rather using the vector of error values, $r_{i}$, such that $|r_{i}|\leq r_{0}$ where i=1,2,…,n and *n* denotes the number of the ODFs (n=50 in our problem). Therefore, our goal was to minimize the maximum absolute error ($r_{0}$) for the mean values of the ODFs. For the second case, the LP problem was defined similarly to solve for the variances. In this problem, the goal was to minimize the maximum absolute error, defined as $e_{0}$, where $|e_{i}|\leq e_{0}$ and i=1,2,…,n and *n* denotes the number of the ODFs or the number of variance parameters (n=50 in our problem). The error vector, *e*, was obtained with the following relation: (16)$$p^{2}\left(\sigma_{A}^{2}+e\right)=\sigma_{S_{11}}^{2}.$$

After definition of the objective functions, we will discuss more details about both problems including the definitions of the equality and inequality constraints in the next section.

### 4.1. LP Problem for ODF Mean Values

The LP problem generated to find the ODF mean values aims to minimize the following objective function:(17)$$\begin{array}{c} \min\;\;r_{0},\\ \mathrm{where}\;\;|r_{i}|\leq r_{0}\;\;i=1,2,\ldots ,n, \end{array}$$
where the variables of the problem are defined as follows: (18)$$\begin{array}{c} x={\left[r_{1},r_{2},\ldots ,r_{n},\mu_{A_{1}},\mu_{A_{2}},\ldots ,\mu_{A_{n}},r_{0}\right]}^{T}, \end{array}$$
where $\mu_{A_{1}},\mu_{A_{2}},\ldots ,\mu_{A_{n}}$ show the mean values of each ODF parameter. Therefore, the total number of variables is found as: 2n+1, which was equal to 101 in our problem. Since the objective of the standard LP problem is to minimize $f^{T}x$, the *f* values can be defined as: (19)$$f={\left[0_{1\times 2n},1\right]}^{T}.$$

The next step is to define the design constraints. The equality constraints in the problem are given as: (20)$$\begin{array}{c} {p_{11}}^{T}\left(\mu_{A}+r\right)=\mu_{S_{11}},\\ q^{T}\mu_{A}=1. \end{array}$$

Note that $q^{T}\mu_{A}=1$ is a strict mathematical constraint to be satisfied. Therefore, it should be satisfied without the tolerance implied by the error parameters. Next, the coefficient matrix of the augmented system of equality equations is defined as:$$\left[P_{aug}\right]=\left[\begin{array}{ccc} p_{11}^{T} & p_{11}^{T} & 0\\ 0_{1\times n} & q^{T} & 0 \end{array}\right],$$
where $0_{1\times n}$ is the vector of zeros with a size of (1×n). The right-hand-side values for the equality constraints are given as: $b_{eq}={\left[\mu_{S_{11}},1\right]}^{T}$. The definition of the objective function through the maximum absolute error ($|r_{i}|\leq r_{0}$) leads to two different inequality constraints (shown in Equation (Equation 21)) to guarantee that $r_{0}$ is always the maximum absolute error:(21)$$\begin{array}{c} -r_{i}-r_{0}\leq 0,\\ r_{i}-r_{0}\leq 0, \end{array}$$
where i=1,2,…,n. The coefficient matrix of the augmented system of inequality constraints is defined as: $$\left[M_{aug}\right]=\left[\begin{array}{ccc} -I_{n\times n} & 0_{n\times n} & -1_{n\times 1}\\ I_{n\times n} & 0_{n\times n} & -1_{n\times 1} \end{array}\right],$$
where $I_{n\times n}$ is the n×n identity matrix, $0_{n\times n}$ is the n×n matrix of zeros, and $1_{n\times 1}$ is the 1×n vector of ones. The right-hand-side values for the inequality constraints are given as: $b={\left[0_{1\times n},1_{n\times n}\right]}^{T}$. Finally, the lower and upper bounds of the design variables are defined. For the lower bounds, we had the following constraints: (22)$$\begin{array}{c} \mu_{A}\geq 0,\\ r_{0}\geq 0, \end{array}$$
where the error vector, *r*, does not have a specific lower limit. The upper bound values were defined only for the ODF mean values, which were set by the single-crystal designs that led to the most extreme mathematically possible solutions.

### 4.2. LP Problem for ODF Variances

The second LP problem was defined to find the variances of the ODFs. The objective function of this LP problem is defined as:(23)$$\begin{array}{c} \min\;\;e_{0},\\ \mathrm{where}\;\;|e_{i}|\leq e_{0}\;\;i=1,2,\ldots ,n, \end{array}$$
where the variables of the problem are defined as follows: (24)$$\begin{array}{c} \begin{array}{c} x={\left[e_{1},e_{2},\ldots ,e_{n},\sigma_{A_{1}}^{2},\sigma_{A_{2}}^{2},\ldots ,\sigma_{A_{n}}^{2},e_{0}\right]}^{T}, \end{array} \end{array}$$
where $\sigma_{A_{1}}^{2},\sigma_{A_{2}}^{2},\ldots ,\sigma_{A_{n}}^{2}$ show the variance of each ODF parameter. Therefore, the total number of variables is found as: 2n+1, which was equal to 101 in our problem. Since the objective of the standard LP problem is to minimize $f^{T}x$, the *f* values can be defined as: (25)$$f={\left[0_{1\times 2n},1\right]}^{T}.$$

The equality constraints in the problem are given as follows: (26)$$\begin{array}{c} p_{11}^{2}\left(\sigma_{A}^{2}+e\right)=\sigma_{S_{11}}^{2},\\ q^{2}\sigma_{A}^{2}=0. \end{array}$$

Note that $q^{2}\sigma_{A}^{2}=0$ is a strict mathematical constraint to be satisfied. Therefore, it should be satisfied without the tolerance implied by the error parameters. Next, the coefficient matrix of the augmented system of equality equations is given:$$\left[P_{aug}\right]=\left[\begin{array}{ccc} p_{11}^{2} & p^{2} & 0\\ 0_{1\times n} & q^{2} & 0 \end{array}\right].$$

The right-hand-side values for the equality constraints are given as: $b_{eq}={\left[\sigma_{S_{11}}^{2},0\right]}^{T}$. The definition of the objective function through the maximum absolute error ($|e_{i}|\leq e_{0}$) leads to two different inequality constraints (shown in Equation (Equation 27)) to guarantee that $e_{0}$ is always the maximum absolute error:(27)$$\begin{array}{c} -e_{i}-e_{0}\leq 0,\\ e_{i}-e_{0}\leq 0, \end{array}$$
where i=1,2,…,n. The coefficient of the augmented system of inequality constraints is defined as:$$\left[M_{aug}\right]=\left[\begin{array}{ccc} -I_{n\times n} & 0_{n\times n} & -1_{n\times 1}\\ I_{n\times n} & 0_{n\times n} & -1_{n\times 1} \end{array}\right].$$

The right-hand-side value vector for the inequality constraints is defined as: $b={\left[0_{1\times n},1_{n\times n}\right]}^{T}$. Finally, the lower and upper bounds of the design variables are defined. For the lower bounds, we have the following constraints: (28)$$\begin{array}{c} \sigma_{A}^{2}\geq 0,\\ e_{0}\geq 0, \end{array}$$
where the error vector, *e*, does not have a specific lower limit.

### 4.3. Finding the Uncertainty in Microstructural Texture

The LP problems for the mean values and variances of the ODFs were solved with the “interior-point” algorithm. The solution obtained with the LP problem was assigned to be the “initial solution” since the underdetermined system can produce multiple solutions, and the LP solution identifies only one of them. The initial solutions (initial mean values and variances) are represented in the Rodrigues orientation space in Figure 3.

The mean values and variances represented in Figure 3 suggest that the initial solution led to a textured design rather than a randomly oriented texture, which is the expected experimental texture. In addition, the ODF values with high mean values also had high variances, as expected. However, multiple solutions can be possible, as the initial solution can be one of the many solutions of an underdetermined system. To explore the multiple solutions, the null space of the solution system should be computed. The null spaces of the mean value and variance systems were identified in separate problems. The following system of equations was considered to find the null space for the mean values: (29)$$\begin{array}{c} p_{11}^{T}\mu_{A}=\mu_{S_{11}},\\ q^{T}\mu_{A}=1. \end{array}$$

Similarly, the following system was considered to find the null space for the variances: (30)$$\begin{array}{c} p_{11}^{2}\sigma_{A}^{2}=\sigma_{S_{11}}^{2},\\ q^{2}\sigma_{A}^{2}=0. \end{array}$$

The multiple solutions for the ODF mean values and variances can be written after identifying the null spaces: (31)$$\begin{array}{c} \mu_{A_{i}}=\mu_{A_{1}}+\lambda V(i),\\ \sigma_{A_{i}}^{2}=\sigma_{A_{1}}^{2}+\beta U(i), \end{array}$$
where $\mu_{A_{i}}$ represents the multiple solutions for the ODF mean values in terms of the initial ODF mean value solution, $\mu_{A_{1}}$ (Figure 3a), and the null space vectors, V(i), of the coefficient matrix. Note that λ can be any real value as long as the ODF non-negativity condition is satisfied (A≥0 or $\mu_{A}\geq 0$ for the ODF mean values). Similarly, $\sigma_{A_{i}}^{2}$ represents the multiple solutions for the ODF variances in terms of the initial ODF variance solution, $\sigma_{A_{1}}^{2}$ (Figure 3b), and the null space vectors, U(i), of the coefficient matrix, and β can be any real number as long as the non-negativity condition for the variances is satisfied ($\sigma_{A}^{2}\geq 0$).

Our main goal was to achieve an analytical scheme for the microstructural uncertainty that can best represent the experimental variations. Therefore, the best matching solution with the experimental data was identified in the next step among all possible solutions. For this purpose, we aimed to find the null space solutions that gave the best match with the experimental data that was available for the ODF mean values and variances. The experimental data were obtained through 150 EBSD image samples for the Ti-7Al alloy, and it was presented in our previous publication [13]. Using these 150 EBSD samples and the direct relationship between the ODF values and EBSD samples (explained in [13]), the mean values and variances of the ODFs were calculated for the experimental data. To find our final microstructure solution, we first considered the following LP problem to minimize the differences between the ODF mean values obtained in the analytical model and experiments: (32)$$\begin{array}{c} \begin{array}{c} \min\;\;d_{0},\\ \mathrm{where}\;\;|d_{i}|\leq d_{0}\;\;i=1,2,\ldots ,n,\\ \mathrm{subject}\;\mathrm{to}:\;\;\lambda V+d=\mu_{A}^{exp}-\mu_{A_{1}},\\ -d_{i}-d_{0}\leq 0,\\ d_{i}-d_{0}\leq 0,\\ q^{T}\left(\mu_{A_{1}}+\lambda V\right)=1,\\ \lambda V\leq \mu_{A_{1}},\\ x={\left[d_{1},d_{2},\ldots ,d_{n},\lambda_{1},\lambda_{2},\ldots ,\lambda_{m},d_{0}\right]}^{T}, \end{array} \end{array}$$
where the objective was the minimization of the maximum absolute error value, $d_{0}$, among all error values, $d_{i}$, which define the difference between the multiple ODF mean value solution, $\mu_{A_{i}}$, and the mean ODF values in the experimental data, $\mu_{A}^{exp}$. The first constraint was also defined in terms of the vector of the error values, *d*. The number of null space vectors was denoted as *m*. The last constraint was derived for the ODF non-negativity condition. Similarly, the LP problem for the ODF variances can be defined as follows: (33)$$\begin{array}{c} \begin{array}{c} \min\;\;g_{0},\\ \mathrm{where}\;\;|g_{i}|\leq g_{0}\;\;i=1,2,\ldots ,n,\\ \mathrm{subject}\;\mathrm{to}:\;\;\beta U+g=\sigma_{A^{exp}}^{2}-\sigma_{A_{1}}^{2},\\ -g_{i}-g_{0}\leq 0,\\ g_{i}-g_{0}\leq 0,\\ q^{2}\left(\sigma_{A_{1}}^{2}+\beta U\right)=0,\\ \beta U\leq \sigma_{A_{1}}^{2},\\ x={\left[g_{1},g_{2},\ldots ,g_{n},\beta_{1},\beta_{2},\ldots ,\beta_{m},g_{0}\right]}^{T}, \end{array} \end{array}$$
where the objective function was defined similarly as the minimization of the maximum absolute error value, $g_{0}$, among all error values, $g_{i}$, which define the difference between the multiple ODF variance solutions, $\sigma_{A_{1}}^{2}$, and the variances in the experimental data, $\sigma_{A^{exp}}^{2}$. The first constraint was also defined in terms of the vector of the error values, *g*. The number of null space vectors was again denoted as *m*. The last constraint was derived for the non-negativity condition for the variances. The solutions of the given LP problems in Equation (Equation 32) for the ODF mean values and Equation (Equation 33) for the ODF variances are shown and compared to the experimental data in Figure 4.

The AUQLin model solution for the ODF uncertainty matched well with the available experimental data for Ti-7Al alloy [13]. Small differences arose since the off-diagonal variance terms are neglected in the AUQLin model solution. However, with the current modeling scheme, the AUQLin was able to capture the mean values and variances of the ODF values in the experimental data set.

## 5. Conclusions

The present work aimed to develop an analytical UQ model—AUQLin—to quantify the uncertainty in the microstructure using the measured variations in Young’s modulus of Ti-7Al alloy. First, the variations in Young’s modulus were captured with the AUQLin model, and an inverse problem was solved to find the uncertainty in the volume-averaged material property, compliance value. The compliance has a linear relationship with the microstructure descriptor, ODF, which measures the volume fractions of different crystallographic orientations in a polycrystalline material. Owing to the linear relationship, the uncertainty in the microstructure was found by solving an inverse problem to obtain the statistical features of the ODF. The inverse problem led to an underdetermined system which produced multiple solutions to the microstructural uncertainty. Next, the experimental data that are available with 150 EBSD samples which were taken from different specimens of the same material, or even from a different location of the same specimen, were used to quantify the uncertainty in the test data. To find the best-matching AUQLin solution with the experimental data, an optimization problem was solved to minimize the difference in the statistical parameters in AUQLin and the available experimental microstructure data. The final solution was found to match well with the available experimental data. 

## Figures and Tables

**Figure 1 materials-12-01773-f001:**
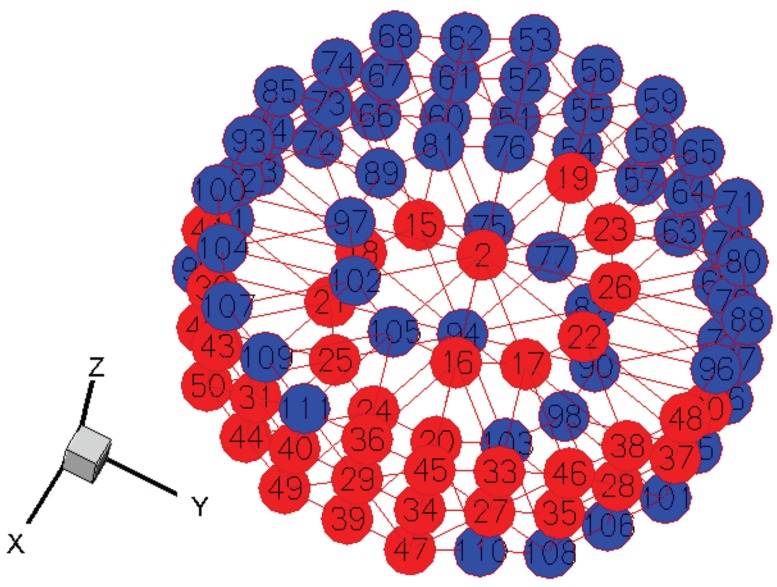
Orientation distribution function (ODF) representation for the hexagonal close-packed (HCP) Ti-7Al microstructure (the independent ODFs are shown in red).

**Figure 2 materials-12-01773-f002:**
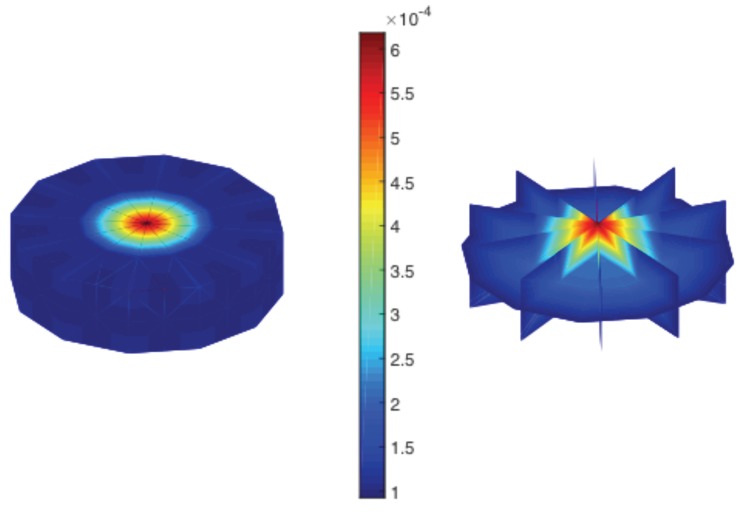
Single-crystal property matrix ($p_{11}$) values in Rodrigues space for the compliance parameter $S_{11}$ (1/MPa).

**Figure 3 materials-12-01773-f003:**
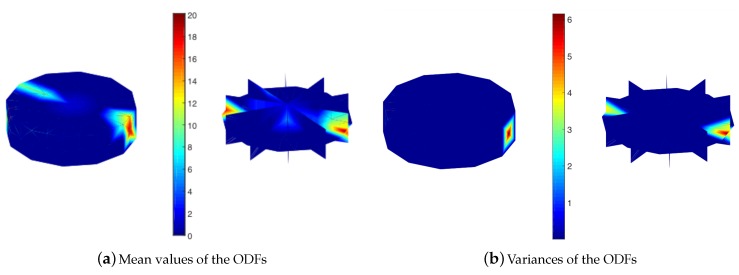
Initial solution for the uncertainty in microstructural texture.

**Figure 4 materials-12-01773-f004:**
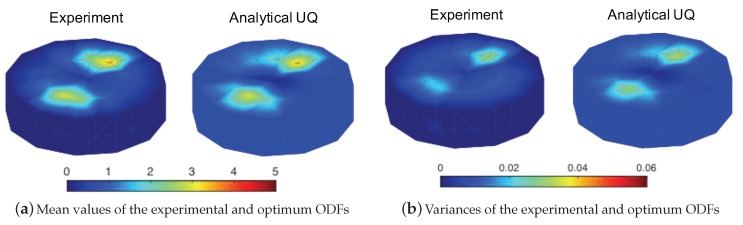
Optimum solution for the uncertainty in microstructural texture. UQ: uncertainty quantification.

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
