# Peer review of "Uncertainty Quantification for Ti-7Al Alloy Microstructure with an Inverse Analytical Model (AUQLin)"

_materials, 2019, doi:10.3390/ma12111773_

Round 1
Reviewer 1 Report
report attached

Author Response
Thank you very much for the review and feedback. Please find enclosed the document that includes our responses to the comments.
Sincerely,

Reviewer 2 Report
The author develop an analtical treatment of uncertainty in microstructure. The present treatment is inevitably limited to the case of linear model and therefore it is of limited interest.
In the literature several attempts to have an insight on the effects of uncertainties for nonlinear models have already been treated, see for example
- A. Tosin, M. Zanella. Boltzmann-type models with uncertain binary interactions. Communications in Mathematical Sciences, 16(4): 962--984, 2018.
- G. Dimarco, L. Pareschi, M. Zanella. Uncertainty quantification in socio economic and life sciences. In Uncertainty Quantification for Hyperbolic and Kinetic Equations, SEMA-SIMAI Springer Series vol. 14, 151-191, 2017.
- J. Hu and S. Jin. Uncertainty quanti cation for kinetic equations. In Uncertainty Quanti cation for Hyperbolic and Kinetic Equations, volume 14, SEMA-SIMAI Springer Series, pages 193-229. Springer, 2017.
In the linear case, furthermore, also the stochastic Galerkin setting is very efficient since it can be easily parallelized.
I also recommend the author to better introduce the topic, in particular in the introduction it is stated that gPC requires a high number of evaluations. To be honest gPC is spectrally accurate therefore it is exacly the opposite, with few projections we reach high accuracy. It is true anyway that SG-gPC methods require the knowledge of the distribution function of the uncertainties, which can be quite strong as an assumption.
Author Response

(The authors gave the same response as above.)

Round 2
Reviewer 1 Report
While improvements have been made, the authors have not answered the key questions about the inability to reduce uncertainty from experimental data in this current setup which presents a fundamental flaw in the framework. Another unanswered question is about that unit volume constraint is only imposed in mean of the distribution.
Therefore my recommendation of rejection remains.
Author Response
Thank you very much for your comments.
Our goal in this paper is to model the variations that are already present in the experimental data rather than minimizing the experimental uncertainty. We develop an analytical model to mimic the variations in material properties using the experimental data, and compute the corresponding uncertainty in the computational microstructure. The uncertainty that we computed for the microstructural texture agrees well with what is already available through experimental measurements of the same microstructure.
The experimental uncertainty is related to aleatoric variations which arise fundamentally from the natural stochasticity of the microstructure and thus it is random.
The unit volume fraction constraint is imposed through both mean value and covariance equations, and the final solution that we obtain for the microstructural texture satisfies the corresponding constraint.
Reviewer 2 Report
The author answered to all my concerns. I recommend the article for publication.
Author Response
Thank you very much for your revisions and for suggesting publication.